# Psoriasis: The Versatility of Mesenchymal Stem Cell and Exosome Therapies

**DOI:** 10.3390/biom14111351

**Published:** 2024-10-24

**Authors:** Aidar Dairov, Aliya Sekenova, Symbat Alimbek, Assiya Nurkina, Miras Shakhatbayev, Venera Kumasheva, Sandugash Kuanysh, Zhansaya Adish, Assel Issabekova, Vyacheslav Ogay

**Affiliations:** 1Stem Cell Laboratory, National Center for Biotechnology, Astana 010000, Kazakhstan or aidardairov@gmail.com (A.D.); a.sekenova@biocenter.kz (A.S.); symbatalimbek@gmail.com (S.A.); nurkina@biocenter.kz (A.N.); shakhatbayev@biocenter.kz (M.S.); kumasheva@biocenter.kz (V.K.); ogay@biocenter.kz (V.O.); 2Department of General Biology and Genomics, L.N. Gumilyov Eurasian National University, Astana 010008, Kazakhstan; 3Obstetrics and Gynecology, Astana Medical University, Astana 010000, Kazakhstan; 4Laboratory of Immunochemistry and Immunobiotechnology, National Center for Biotechnology, Astana 010000, Kazakhstan; zhansaya.adish@mail.ru; 5Department of Natural Sciences, L.N. Gumilyov Eurasian National University, Astana 010008, Kazakhstan

**Keywords:** psoriasis, mesenchymal stem cell, mesenchymal-stem-cell-derived exosome, cell therapy, immunomodulation, preconditioning, in vitro, in vivo, clinical study

## Abstract

Mesenchymal stem cells (MSCs) are multilineage differentiating stromal cells with extensive immunomodulatory and anti-inflammatory properties. MSC-based therapy is widely used in the treatment of various pathologies, including bone and cartilage diseases, cardiac ischemia, diabetes, and neurological disorders. Along with MSCs, it is promising to study the therapeutic properties of exosomes derived from MSCs (MSC-Exo). A number of studies report that the therapeutic properties of MSC-Exo are superior to those of MSCs. In particular, MSC-Exo are used for tissue regeneration in various diseases, such as healing of skin wounds, cancer, coronary heart disease, lung injury, liver fibrosis, and neurological, autoimmune, and inflammatory diseases. In this regard, it is not surprising that the scientific community is interested in studying the therapeutic properties of MSCs and MSC-Exo in the treatment of psoriasis. This review summarizes the recent advancements from preclinical and clinical studies of MSCs and MSC-Exo in the treatment of psoriasis, and it also discusses their mechanisms of therapeutic action involved in the treatment of this disease.

## 1. Introduction

Psoriasis is one of the most common clinically heterogeneous, immune-mediated, inflammatory, lifelong skin diseases [1,2]. Psoriasis is characterized by hyperproliferative keratinocytes and infiltration of T cells, dendritic cells (DCs), macrophages, and neutrophils [1]. The etiopathology of psoriasis is not yet clear but it seems that genetic, immunological, and environmental factors are involved [3,4]. Approximately 125 million people worldwide suffer from psoriasis [5,6]. Thirty percent of individuals with psoriasis experience joint problems in addition to skin symptoms [6,7]. People of European heritage and those living in Western nations are primarily affected with psoriasis. Norway (4.6%), France (4.42%), Portugal (4.4%), the United States of America (3.0%), the United Kingdom (2.8%), Germany (2.78%), Canada (2.44%), and other nations distant from the equator are among those with a high prevalence of psoriasis [8].

Mesenchymal stem cells (MSCs) are multilineage-differentiating stromal cells (adipocytes, osteocytes, chondrocytes) with the capacity to self-renew [9]. MSCs have extensive immune-modulatory and anti-inflammatory capabilities [10]. MSCs can be isolate from many tissues, including the umbilical cord, endometrial polyps, menstrual blood, bone marrow, adipose tissue, etc. [9]. MSCs are currently the most common cell type in regenerative medicine. Numerous studies have demonstrated the potential benefits of MSC-based therapies for the treatment of various pathologies, such as bone and cartilage diseases, cardiac ischemia, diabetes, and neurological disorders [11]. In addition to numerous advantages and benefits, there are a number of problems limiting the widespread use of MSC therapy in clinical practice, which include the risk of developing tumors and transmitting viruses and prions after stem cell transplantation; how long-term in vitro cultivation of MSCs leads to loss of the potential of MSCs for differentiation and morphological changes and also increases probability of malignant transformation; low survival and engraftment rates of MSCs due to their short-lived viability after injection; the low therapeutic effect and increased immunogenicity in differentiated MSCs; the heterogeneity of MSCs due to differences in the health status, genetics, gender, and age of donors; varying degrees of stem cell stability and differentiation capacity between MSCs isolated from different sources; different levels of expansion ability under different culture conditions; immune compatibility between donors and recipients; ethical issues; and high manufacturing costs [12,13,14,15]. Consequently, long-term research and monitoring are required to examine the long-term effects of MSC therapy, including any negative effects [12].

Along with MSCs, the therapeutic effects of their exosomes (MSC-Exo) are being extensively investigated. This is mostly because stem-cell-derived exosomes have various advantages over stem cells, including non-immunogenicity, non-infusion toxicity, easy access, easy storage, and the absence of tumorigenic potential and ethical problems [16]. Exosomes are lipid bilayer vesicles that are spherical and have a diameter that varies from 30 to 150 nm. Exosomes exhibit selective enrichment for distinct biomolecules, such as membrane glycoproteins, lipids, and other cell-specific proteins, in addition to multiple types of nucleic acids [17]. Exosomes are involved in intercellular communication through a variety of cargo types and can influence the immune response by interacting with immune effector cells in the presence of anti-inflammatory compounds [18]. MSC-Exo are used to mediate tissue regeneration in a variety of diseases, such as cutaneous wound healing [19], cancer [20,21,22], ischemic heart disease, lung diseases [23], liver fibrosis, and neurological, autoimmune, and inflammatory diseases [18]. As already mentioned, a significant advantage of MSC-Exo compared to MSCs is that they overcome the limitations of cell therapy by providing comparable benefits in a safer and more stable extracellular vesicle format. Moreover, the contents of exosomes can also be modified to enhance regenerative biological activity [24].

In this regard, it is not surprising that the scientific community is interested in studying the therapeutic properties of MSCs and MSC-Exo in the treatment of psoriasis. This review summarizes the recent advancements from preclinical and clinical studies of MSCs and MSC-Exo in the treatment of psoriasis, and it also discusses their mechanisms of therapeutic action involved in the treatment of this disease.

## 2. Disease Pathogenesis Mechanism

Psoriasis pathogenesis is governed by the interleukin (IL)-23/IL-17 signaling axis [25]. This complex signaling mechanism involves members of both the innate and adaptive immune systems. Disease onset begins with immune activation in genetically predisposed individuals following environmental triggers such as infection, medication, and smoking [26]. Another concomitant event triggering disease initiation is the loss of immune tolerance through the recognition of autoantigens, specifically antimicrobial peptides like LL-37/cathelicidin released by keratinocytes [27]. Individuals with genetic susceptibility also release self-nucleotides that can form complexes with LL-37 and are recognized by toll-like receptors (TLR7 and TLR9) on the surface of plasmacytoid dendritic cells (pDCs) [28,29]. This binding event activates pDCs, eliciting the secretion of inflammatory mediators interferon-α (IFN-α) and IFN-β to stimulate other dermal DC subsets to produce proinflammatory mediators, such as the primary cytokine IL-23, IL-12, and tumor necrosis factor (TNF) [30]. Activated pDCs and other DC subsets present the psoriatic autoantigen LL-37/cathelicidin to CD4+ and CD8+ T cells. Antigen presentation can occur within the dermis, stimulating resident memory T cells, and in the draining lymph nodes, where it activates naïve T cells. At the same time, secreted IL-23 evokes the further activation and clonal expansion of IL-17- and IL-22-secreting T helper (Th)17 and Th22 cells, respectively [31].

Active Th17 cells exert their downstream effect through several cytokines IL-17, IL-26, IL-29, and TNF-α. They play a significant role in creating a feed-forward loop that exacerbates the disease state by recruiting other cell types. First, their key cytokine IL-17 targets keratinocytes that express IL-17 receptors, inducing the expression of CC-chemokine ligand 20 (CCL20) [32]. This chemokine attracts IL-23-producing DCs and Th17 cells, further compounding the already inflamed environment. IL-17 drives disease pathogenesis by activating psoriasis-related genes in keratinocytes via IL-22, IL-19, and IL-36 to increase epidermal hyperplasia and produce more antimicrobial peptides, including LL-37/cathelicidin [33]. IL-17 also promotes and maintains an inflammatory environment by attracting additional innate immune cell populations. In particular, circulating neutrophils aggregate at the inflamed site due to the release of neutrophil-attracting factors such as chemokine (C-X-C motif) ligand (CXCL)1/2/3/5 and CXCL8 [34].

While the IL-23/IL-17 signaling axis is pivotal in disease onset and progression, recent studies have explored the active involvement of neutrophils in psoriasis pathogenesis. Neutrophils employ a distinct mechanism called NETosis to eliminate foreign bodies. NETosis is a form of cell death in which neutrophil extracellular traps (NETs)—web-like structures composed of cytosolic proteins and decondensed DNA/RNA—are released into the surrounding environment [35,36]. Upon receiving inflammatory stimuli, recruited neutrophils actively form NETs within psoriatic lesions. Notably, the severity of psoriasis correlates with the quantity of NETs in blood samples [37]. These NETs are abundant in LL-37 and RNA. LL-37 can form complexes with RNA, facilitating RNA’s uptake by neutrophils. This process subsequently activates TLRs and leads to the secretion of IL-8, a neutrophil chemotactic factor that recruits additional neutrophils to the lesion site [38]. Intriguingly, the same complex can induce neutrophils to release more NETs, thereby propagating the inflammatory cycle [39].

Thus, in the IL-23/IL-17 disease model, dermal DCs release IL-23, which eventually induces Th17 cell activation and proliferation. Th17 cells produce proinflammatory cytokines to target keratinocytes, which sustains and enhances the chronic inflammatory state by generating additional IL-23, as well as other proinflammatory cytokines, chemokines, S100 family proteins, and antimicrobial peptides. This repetitive cycle perseveres and amplifies the ongoing inflammatory psoriatic process.

## 3. Signs and Symptoms of Psoriasis

Diagnosis of psoriasis in a population through a definite procedure by noting signs and symptoms is meaningful for determining the method of treatment. Previously, researchers confirmed that the application of the Psoriasis Symptom Inventory (PSI) by patients themselves, along with Static Physician Global Assessment (sPGA) and Body Surface Area (BSA), were meaningful for diagnosing psoriasis and the subsequent choice of treatment. Also, psoriasis symptoms of the median degree were found to actually take a severe form in patients [40].

The authors defined such signs of psoriasis as (1) damage to the distal interphalangeal joints; (2) axial damage to three joints of one finger; (3) early involvement of the toes; (4) thalalgia; (5) the presence of skin rashes; (6) psoriasis in the family; (7) negative rheumatoid factor; (8) osteolysis; (9) sacroiliitis; and (10) development of paravertebral ossifications. Within this, the symptoms were characterized as asymmetrical damage in the joints; purple-bluish coloration of the skin over the affected joint, pain and swelling of the joints, and early damage to the big toe; mild muscle atrophy near the affected joints, and articular syndrome (swelling, pain on palpation, changes in the nail plates) [41]. Patients with psoriasis may also have the following symptoms, with effects on the quality of life: psychological disorder, past acuteness, alexithymia, anxiety, and depression. It was noted that the use of the questioning method was very useful for defining those symptoms [42]. Korman et al. defined the next most common signs of psoriasis as erythematous occurrence and the formation of apathetic, flaking plaques of the skin [43].

The risk factors for psoriasis include air pollution, drug intolerance, vaccination, infections, smoking, alcohol consumption, metabolic syndrome, obesity, diabetes, dyslipidemia, hypertension, and mental stress [3]. Also, suffering from psoriasis can lead to the following consequences: systemic inflammation and cardiovascular diseases, metabolic syndrome, hypertension, high lipid levels, liver diseases, diabetes, obesity, deterioration in the quality of life, anxiety, sleep contravention, decrease in cognitive functions, and depression [3,44,45,46].

Genetic factors play a substantial role in the onset and progression of psoriasis. Certain populations have genetic predispositions that make them more susceptible to developing the condition. Previous genome-wide studies identified several major susceptibility loci. Among them, the major histocompatibility complex (MHC) region, particularly class I HLA genes like HLA-C, is strongly associated with psoriasis. The HLA-C*06:02 allele is especially significant, being identified as a primary genetic factor in psoriasis susceptibility across different populations [47].

Genetically predisposed individuals are subject to both external and internal factors that elicit psoriasis initiation and pathogenesis. External risk factors include mechanical damage, individual habits, such as smoking and alcohol consumption, as well as environmental factors like air pollution and sun exposure [48,49,50].

Psoriasis patients may suffer from various infections due to their weakened immune system. Accordingly, patients who are administered vaccinations against infections experience further exacerbation of their condition. For instance, studies pointed out the possible association between influenza vaccine and psoriasis onset [51,52]. Other researchers found that patients developed psoriasis post BCG (Bacillus Calmette–Guerin) vaccine shots [53].

Beyond external triggers, internal factors exacerbate the perpetuation of the condition. For instance, people with other medical conditions, including diabetes mellitus, dyslipidemia, and obesity, are at a higher risk of developing psoriasis [54,55,56]. Hypertension and mental stress are other risk factors, and the latter needs to be extensively investigated to further corroborate its relationship with psoriasis progression. Hypertension, in contrast, was considerably associated with an increased risk of psoriasis [57].

## 4. The Diagnosis of Psoriasis

At the first stage of the diagnosis of psoriasis, patients undergo a viewing and evaluation of the skin and mucosal surfaces (presence of whitish scales and “red dots”), give samples to be tested for fungal infections, and submit to an examination of the whole body, limbs, and nails. The presence of a hereditary skin condition is usually taken into consideration. In turn, clinical diagnostic methods include video dermoscopy (dermatophytes), tests for bacterial infections (corynebacteria, staphylococci, etc.), and reflectance confocal microscopy of skin biopsies (assessing the presence of melanocytic lesions, skin tumors, and some inflammatory and infectious diseases) [58,59]. The standard diagnosis of psoriasis includes skin assessment, disease history, family history, histopathology, and dermoscopy. Diagnosis of psoriasis using Traditional Chinese Medicine (TCM) was based on symptoms of blood fever, blood tension, blood dehydration, and fiery blood heat-toxicity [46].

Classifying the severity of plaque psoriasis can particularly help in choosing the appropriate treatment. Commonly used tools for grading plaque psoriasis by clinicians are the Psoriasis Area and Severity Index (PASI), BSA, and Dermatologic Quality of Life Index (DLQI) [46,60]. Psoriasis diagnosing using dermoscopy is a favored method for evaluating the degree of severity and prognosis of the treatment effectiveness in the clinical regard [61].

The advanced method of using the “ResNet-34 model” was reported, which represents an extensive opportunity for the diagnosis of psoriasis [62]. Another method was analyzing saliva using a linear discriminant analysis as the method for measuring the level of dermatopathies in patients. The efficiency of the method was approximately 87.5% compared to a support vector machine [63]. Other authors found that using the combination of reflectance confocal microscopy and dermoscopy methods was more effective for psoriasis diagnosis [64]. Another new method is a gene expression method in the formalin-fixed and paraffin-embedded tissues of patients’ skin samples, which showed moderate efficiency as a molecular test for diagnosing psoriasis. The authors developed the formalin-fixed and paraffin-embedded-based molecular classifier with a sensitivity/specificity of 92/100% [65]. It was noted that the development of specific markers enhances the diagnosis of generalized pustular psoriasis (GPP). Thus, the application of genetic testing with a medical history, physical examination, laboratory tests, and histopathologic evaluation presents a scenario where the full information is taken into account for the correct diagnosis of GPP [66]. It was noted that with progress in the diagnosis of skin diseases, the recognized methods such as dermoscopy, which have high value, should not be forgotten. For the diagnosis of nail psoriasis, dermoscopy provides all necessary information on the severity of the disease [67]. Therefore, patch testing showed efficiency in distinguishing psoriasis from acute contact dermatitis [68]. Also, Yu et al. found that palmoplantar psoriasis can be diagnosed if the patient gives an indication of white scales, dots, and annular eruption on the skin. However, as reported, yellow-labeled scales and abnormal vessel allocation are typical of palmoplantar eczema [69]. Analysis of scratch psoriasis by the typical clustering algorithm showed that “K-means” and “Fuzzy C-means” with seeker optimization had high reliability [70]. Controlling the levels of 25-(OH) D3, Cu/Zn, and high-density lipoprotein cholesterol (HDL-C) is very informative for the diagnosis of psoriasis [71]. While exploring the level of psoriasis-specific biomarkers comes in handy, the routinely used diagnosis methods such as physical evaluation and PASI are still proven to be useful [72]. Ibad et al. suggested using the terms “psoriasiform” and “spongiotic psoriasiform” when diagnosing psoriasis [73].

Along with standard methods of diagnosis, there are other non-invasive techniques that make use of imaging along with the dermoscopy method; such other methods include conventional ultrasound, high-frequency ultrasonography (HFUS), optical microangiography (OMAG), laser Doppler imaging (LDI), multiphoton tomography (MPT), video-capillaroscopy (VC), reflectance confocal microscopy (RCM), and optical coherence tomography (OCT) [74]. Previosly, authors reported that videodermoscopy is more significant for accurate vessel analysis after dermoscopy analysis while HFUS can be considered for therapy outcome examination [74]. The technological progress made in the diagnosis of psoriasis contributes to the application of modern techniques such as a photoacoustic imaging. The application of ultra-broadband optoacoustic mesoscopy allows for measuring the epithelial thickness, mean diameter of the vessels, and fluorescently labeled psoriasis markers directly in the human body [75].

Previous, researchers reported that measuring the inflammatory cytokines, immunoglobulin A, and antioxidants in the gingival crevicular fluid could be considered, as these are specific biomarkers in psoriasis. Furthermore, assessing the levels of S100A8, IL-18, and sE-selectin in saliva can also be useful, as these are potential biomarkers for diagnosing psoriasis [76]. Interestingly, it was found that the signal transducer and activator of transcription 2 (STAT2) and caspase 3 could be used as a biomarker of psoriasis, because its advanced expression was distinguished as an obvious feature of psoriasis [77].

Histological diagnosis based on samples from patients with psoriasis was characterized by the existence of typical acanthosis, thickening of the epidermis, mitoses above the basal layer, parakeratosis, a decreased density of the granular layer, extended blood vessels, and the presence of neutrophils in the epidermis. Measuring the Ki-67 level in comparison with cyclin D1 can also be helpful for the diagnosis of psoriasis. Moreover, a definite diagnosis of psoriasis can be made by analyzing the nitric oxide synthase 2/inducible nitric oxide synthase gene expression in histological samples [73]. In the histological analysis of psoriasis, the total dissolution and segregation of the laminar parakeratotic layer corneum and enlarged cells with the structured lengthening of the mesh rims coincides with the sign ñ on biopsy samples from histopathology [78].

## 5. Treatment Methods for Psoriasis

The decision on the treatment methods should be made according to standard guidelines, applied and personalized considering the disease history of the patient [46]. Depending on the severity of the disease, the potential treatment approaches include systemic therapy by biologics, namely with monoclonal antibodies, which are IL-17, TNF-α, IL-12, IL-23 inhibitors (e.g., infliximab, ixekizumab, secukinumab, apremilast), or corticosteroid injections; oral therapy by immunosuppressants (methotrexate, cyclosporine A, etc.); topical therapy by surface preparations (creams containing steroid hormones, vitamin D3, etc.); phototherapy (irradiation with ultraviolet light); and selective toxins to lymphocytes (DAB389IL-2) [46,79,80,81,82,83,84,85]. Also, cell therapy an offer an effective and safe way of treating the disease. In particular, the application of MSCs and MSC-derived exosomes plays a crucial role in modulating inflammation and tissue repair in psoriasis [86,87,88,89,90].

General information about the types of psoriasis, risk factors that cause this disease, treatment methods, and their side effects is summarized in Table 1 and Figure 1.

Other kinds of treatments, from Traditional Chinese Medicine, are topical applications, herb packs, baths with medication, fumigation, acupuncture, cupping, fire needles, and auricular acupuncture [46]. However, the success of their application in patients is subject to questions.

Depending on the treatment type, patients may experience a wide range of side effects. For instance, those who receive topical medications experience minimal side effects, such as skin thinning and irritation. Phototherapy often brings about more side effects, including dry, wrinkled, and inflamed skin, with some even developing an increased risk of skin cancer. People administered biologics and oral medications experience common side effects, including muscle soreness, an upset stomach, and loss of appetite, and extending to a suppressed immune system [82] (Figure 1).

## 6. Immunomodulatory Capability of MSCs and the Role of MSC Preconditioning

The MSC is a type of adult stem cell with immunosuppressive and anti-inflammatory properties [96]. It can be used in cell therapy due to its differentiation potential and low immunogenicity [97]. MSCs can differentiate into various specialized cell types, including adipocytes, chondrocytes, osteoblasts, endothelial cells, and cardiomyocytes [98].

The immunomodulatory properties of MSCs are due to their effects on the cytokine profile of lymphocytes [99]. They also inhibit the activity of certain immune cell types such as neutrophils, DCs, anti-inflammatory macrophages, natural killer (NK) cells, and T and B cells, contributing to an overall immunosuppressive environment [100]. The immunomodulatory effects of MSCs are achieved through the paracrine production of various growth factors, cytokines, and mediators of angiogenesis that enable them to modulate immune responses both directly and indirectly. The secretion of IL-8, IL-6, and granulocyte-macrophage colony-stimulating factor (GM-CSF) by MSCs enhances neutrophil migration to the infection or injury, thereby increasing their activation and phagocytic capacity, as well as promoting their survival. Additionally, MSCs produce a number of important molecules such as transforming growth factor-beta (TGF-β), hepatocyte growth factor (HGF), prostaglandin E2 (PGE2), soluble human leukocyte antigen-G5 (HLA-G5) protein, indoleamine-2,3-dioxygenase (IDO), induced nitric oxide synthase (iNOS), and IL-10, IL-4, and IL-2 to influence the proliferation, differentiation, maturation, and functional activity of various immune cell types, such as T cells, B cells, DCs, macrophages, or NK cells. Furthermore, MSCs do not express MHC II, CD40, CD80, and CD86 molecules, which play a role in T-cell activation during graft rejection [101]. MSCs express intercellular adhesion molecule-1 (ICAM-1), which can interact with lymphocyte function-associated antigen-1 (LFA-1) on T cells and NK cells, thereby reducing their cytotoxicity and proliferation [102,103]. MSCs can express non-classical human leukocyte antigen-G (HLA-G), a molecule that plays a role in immunological tolerance. HLA-G expression by MSCs can inhibit the cytotoxic activity of NK cells and cytotoxic T cells and increase the expansion of regulatory T cells (Tregs) [104].

One of the main mechanisms by which human MSCs exert immunosuppressive effects is the production of IDO, which is involved in the degradation of L-tryptophan, leading to its depletion and the accumulation of kynurenine. This in turn inhibits the activation, proliferation, and functional activity of T cells, DCs, NK cells, and Th17 cells’ differentiation. Inflammatory diseases stimulate MSC activation via the action of certain cytokines such as IFN-γ, TNF-α, IL-17, and IL-1β. This activation leads to increased expression of MHC class I/II molecules and costimulatory molecules. It also increases the proliferation and survival of these cells and their immunomodulatory and immunosuppressive functions [105].

Therapeutic and immunomodulatory effects of cytokines, growth factors, enzymes, and other molecules are partially derived from the activity of MSCs and partially from their interaction with the inflammatory environment. MSCs can produce various cytokines, including TGF-β, HGF, and PGE2, especially in response to inflammatory stimuli such as interferon-gamma (IFN-γ) and TNF-α. Additionally, other cytokines like IL-6, IL-10, and IL-1β are often found in the environment where MSCs are active, and their production can be further enhanced through the crosstalk between MSCs and immune cells. This dynamic interaction between MSCs and the inflammatory microenvironment ensures a coordinated production of these molecules, which contribute to the modulation of immune responses and tissue repair processes [106].

The therapeutic and immunomodulatory effects of cytokines, growth factors, enzymes, and other molecules produced by MSCs are summarized in Table 2.

Preconditioning with IFN-γ or TNF-α can increase IDO synthesis in MSCs. Subsequently, this promotes the differentiation of monocytes into IL-10-secreting M2 macrophages. These M2 macrophages exert immunosuppressive effects. Pretreatment of MSCs with IL-17 inhibits the secretion of T helper 1 (Th1) cell cytokines (TNF-α, IFN-γ, and IL-2) by T cells and also promotes the activation of Tregs [108]. Additionally, molecules like ICAM-1 can be upregulated in the presence of an inflammatory microenvironment, thereby strengthening the interaction between MSCs and immune cells [102].

MSC-Exo are key mediators of their immunomodulatory effects. Exosomes contain microRNAs (miRNAs), proteins, and lipids that can alter the activity of immune cells, such as reducing the expression of proinflammatory cytokines in macrophages and promoting their conversion to an anti-inflammatory phenotype. Through the release of exosomes containing immunoregulatory molecules, MSCs can inhibit the activity of proinflammatory immune cells and support the function of Tregs [122].

In summary, the immunomodulatory effects of MSCs are the result of both their intrinsic properties and their dynamic response to external inflammatory cues, as well as their ability to influence immune responses through both secreted factors and direct cellular interactions [106].

## 7. Preclinical Studies

### 7.1. In Vitro Studies

In vitro studies investigating the effects of MSCs on psoriasis offer intricate insights into the pathophysiology of this complex dermatological disorder. In the context of psoriasis, the microenvironment exhibits distinct characteristics compared to atopic dermatitis and healthy skin. This specific environment induces resident MSCs to release proinflammatory and angiogenic mediators, while simultaneously diminishing their antioxidant capacity, a combination that contributes to the development of psoriatic skin lesions. This highlights the importance of understanding the dynamic interplay between the microenvironment and resident MSC populations in disease progression [123].

Furthermore, psoriatic dermal MSCs (p-DMSCs) play a multifaceted role in the pathogenesis of psoriasis. These cells display a remarkable ability to upregulate the expression of vascular endothelial growth factor (VEGF), thereby promoting angiogenesis and facilitating the migration of human umbilical vein endothelial cells (HUVECs). This angiogenic potential suggests a contributory role of p-DMSCs in the enhanced vascularity observed in psoriatic lesions [124]. Additionally, p-DMSCs exhibit a dual effect on keratinocytes, the predominant cell type in the epidermis. They not only promote keratinocyte proliferation but also suppress apoptosis by downregulating caspase-3 expression. Consequently, this dysregulation in keratinocyte dynamics leads to hyperproliferation and impaired apoptosis, exacerbating the inflammatory response within the epidermal layer [125].

Moreover, normal dermal MSCs (n-DMSCs) have been shown to outperform p-DMSCs in suppressing T-cell proliferation and enhancing Treg activity. It was found that p-DMSCs have a higher expression of genes responsible for the activation of Th1 and Th17 cytokines than n-DMSCs. Compared to p-DMSCs, n-DMSCs not only more effectively inhibit CD3 T-cell proliferation but also promote T-cell apoptosis, leading to a reduction in the Th1/Treg ratio. This suggests that n-DMSCs could partially restore the immune balance through upregulation of TGF-β receptor signaling pathways, enhancing the immunosuppressive function of Tregs [126,127].

Finally, a study aimed to evaluate the influence of hUC-MSCs on MSCs derived from psoriatic patients (PsO-MSCs), and it shed light on the interaction between different MSC populations. hUC-MSCs and PsO-MSCs were isolated and characterized, followed by an indirect co-culture experiment. The effects of co-culturing on the proliferation and expression of cytokines associated with Th1/Th17 and Th2 pathways were assessed. Intriguingly, the results revealed that prior to co-culturing, the proliferation of PsO-MSCs was significantly higher than that of hUC-MSCs, suggesting inherent differences in the proliferative capacities of MSCs derived from psoriatic patients compared to those from healthy sources [128].

Moreover, both unmodified MSCs and those treated with superoxide dismutase 3 (SOD3) exhibit profound inhibition of T-cell proliferative responses in vitro. This inhibition underscores the potential of MSC-based therapies in modulating the dysregulated immune responses characteristic of psoriasis, potentially offering a novel avenue for therapeutic intervention [129]. Furthermore, recent research highlights that exosomes derived from IFN-γ-stimulated hUC-MSCs exhibit notable immunomodulatory effects. These vesicles, by inhibiting peripheral blood mononuclear cell and T-cell proliferation, significantly reduced cytokine production linked to inflammation in vitro [130].

One compelling finding is the capacity of human umbilical cord MSC-derived exosomes (hUCMSCs-Exo) to suppress DCs’ maturation and activation. This suppression is accompanied by a significant reduction in the expression levels of IL23, a pivotal cytokine implicated in the initiation and perpetuation of psoriatic inflammation [131].

In another study, human umbilical cord blood mononuclear cell-derived exosomes (UCB-MNC-Exo) decreased the expansion of CD4+ and CD8+ T cells and the production of cytokines, and they promoted an increase in Treg levels in vitro. UCB-MNC-Exo lowered levels of IL-6, IL-8, C-X-C motif chemokine ligand 10 (CXCL10), cyclooxygenase-2 (COX-2), S100A7, and defensin beta 4 (DEFB4), which are markers associated with inflammation and the psoriatic disease process [132].

Overall, these comprehensive in vitro findings delineate the specific interplay between MSCs and various cellular components involved in psoriasis pathogenesis. Such insights provide a solid foundation for the development of targeted therapeutic strategies aimed at ameliorating the symptoms and halting the progression of psoriasis.

### 7.2. In Vivo Studies

MSCs and MSC-Exo alleviate psoriasis symptoms through a variety of molecular mechanisms. One key mechanism involves the enhancement of CD4+ T cell activity, which leads to an increased production of IL-4+ and induces apoptosis in cytotoxic T cells. Additionally, MSCs and their exosomes modulate the function of Th1, Th2, Th17, and Treg cells, resulting in elevated levels of IL-10. These actions help maintain an anti-inflammatory environment. The inflammatory environment activates keratinocyte proliferation, which further produces inflammatory cytokines (IL-17A, IL-22) and chemokines (CCL20, CCL27). This positive feedback loop could be halted via the inhibitory action of MSCs and their exosomes. Furthermore, these cells can target dendritic cells, macrophages, and other T cells to curb the production of additional inflammatory factors [133].

At the same time, preclinical studies in animal models (in vivo) have reported new mechanisms of action of MSCs and MSC-Exo that contribute their therapeutic effect in alleviating the symptoms of psoriasis. These mechanisms of action are summarized in Figure 2 and are also described below in the text.

Sah, et al. found that subcutaneous injection of allogeneic SOD3-transduced MSCs significantly prevented psoriasis development in imiquimod (IMQ)-induced psoriasis-like skin inflammation in mice. They identified that it happens through a suppression of proliferation and infiltration of various effector cells into skin with a concomitant modulated cytokine and chemokine expression and inhibition of signaling pathways such as toll-like receptor-7 (TLR-7), nuclear factor-kappa B (NF-*k*B), p38 mitogen-activated protein (MAP) kinase, and Janus kinase–signal transducer and activator of transcription (JAK-STAT), as well as adenosine receptor activation [129].

In another study, Lee, et al. showed that subcutaneous injection of MSCs prevented and treated IMQ-induced and IL-23-mediated psoriasis-like skin inflammation in a mouse model. The mechanism of action of the MSC is the inhibition of the expression of proinflammatory cytokines (IL-6, IL-17, TNF-α) and chemokines (CCL17, CCL20, CCL27) [134].

Kim, et al. demonstrated that transplantation of tonsil-derived MSCs (T-MSCs) into IMQ-induced psoriasis-like skin inflammation in mice significantly abrogated disease symptoms. This occurs primarily by blunting the Th17 response in a PD-L1-dependent manner, resulting in decreased gene expression of IL-23, TNF-α, IFN-γ, IL-17, IL-22, K6, K16, and CCL20 [135].

In a report of Chen, et al., in a mouse model of IMQ-induced psoriasis-like skin inflammation MSCs significantly reduced the expression level of proinflammatory cytokines (IL-17, IL-23, IL-6, and IL-1β) and keratinocyte differentiation markers (S100A7, S100A8, and S100A9), and they remarkably increased the expression level of anti-inflammatory cytokine IL-10 [136].

Imai, et al. identified that in mice treated with human amnion-derived MSCs (hAMSC), the gene expression levels of IL-17A, IL-22, and CXCL1 were significantly reduced in the mouse model of IMQ-induced psoriasis [137].

In another study, researchers studied the effect of topical application of MSC-Exo in a mouse model of IMQ-induced psoriasis. They established that MSC-Exo significantly reduce the content of proinflammatory cytokines IL-17 and IL-23 and also terminal complement activation complex C5b-9 in psoriatic skin. According to their suggestion, topical application of MSC-Exo inhibits C5b-9 activation through CD59 in the stratum corneum. Thus, it alleviates IL-17 release by neutrophil extracellular traps (NET_S_) from neutrophils that accumulate in and beneath the stratum corneum [138].

Zhang, et al. in a recent study reported that hUCMSCs-Exo could inhibit psoriasis-like skin inflammation by suppressing the expression of IL-17, IL-23, and CCL20, thereby inhibiting the phosphorylation of signal transducer and activator of transcription 3 (STAT3) [131].

Rokunohe et al. in a mouse model of psoriasis demonstrated that local adipose-derived stromal cells’ (ASCs’) application inhibited the IMQ-induced upregulation of IL-17A and TNF-α expression and maintained a clinically normal environment in murine skin [139].

An interesting study was conducted Xu, et al., in which they studied MSC-Exo with high PD-L1 expression (MSC-Exo-PD-L1) for the treatment of psoriasis in a mouse model of IMQ-induced psoriasis. MSC-Exo-PD-L1 significantly suppressed the inflammatory response via a reduction in immune cell infiltration, alteration of their phenotype, activation of immunoregulatory, cells and regulation of inflammatory and immunoregulatory cytokines in skin and the peripheral circulation, which was broken by PD-L1 antibody treatment. They showed that MSC-Exo-PD-L1 restored tissue lesions by inhibiting inflammatory immune cells via the PD-1/PD-L1 pathway [140].

Chen et al. discovered that subcutaneous administration of hUC-MSCs drastically diminished the severity of IMQ-induced psoriasis-like dermatitis and suppressed the inflammatory cell response. Furthermore, they revealed that hUC-MSCs may repress skin inflammation, probably by inhibiting interleukin-17-producing γδ T cells [86].

In a recent study, Ren et al. investigated the matrix metalloproteinase-13 (MMP13) that plays a key role in extracellular matrix (ECM) remodeling. They found that MMP13 was upregulated in the skin lesions of an IMQ-induced mouse model, and it was downregulated after intravenous infusion of hUC-MSCs. The researchers suggested that systematically infused hUC-MSCs exert a therapeutic effect on psoriasis through the TNF-α/NF-κB/MMP13 pathway [141].

In another recent study, Cuesta-Gomez et al. compared the efficacy of an infusion of AD-MSCs with the application of BM-MSCs in a mouse model of IMQ-induced psoriasis-like skin inflammation. They described that improved recovery of the skin was associated with increased IL-17A and TGF-β in the skin of mice treated with BM- or AD-MSCs, and they hypothesized that TGF-β promoted the controlled differentiation of keratinocytes, resulting in the decreased severity of psoriasis [142].

Rodrigues et al. reported that administration of UCB-MNC-Exo increases the number of Tregs in the skin and prevents acanthosis in imiquimod-induced psoriasis without affecting the overall disease burden [132].

Another group of researchers, Attia et al., investigated the possible effect of hUC-MSCs when compared with conventional betamethasone cream treatment on IMQ-induced psoriasis-like skin lesion in a rat model. In their study, MSCs demonstrated efficacy in reducing disease severity. Psoriatic symptoms, high expression of inflammatory mediators, and immune cell infiltration into the skin were alleviated after MSC administration. According to the research team, the mechanisms of action of MSCs are the regulation of immune cell infiltration, especially Th17 cells, and the regulation of epidermal functions and differentiation [143].

Thus, it can generally be said that a sufficiently large array of data on the application and the possible anti- and immunomodulatory effects of MSCs and MSC-Exo in psoriasis are being studied. The existing research data differ from each other due to various specific variables employed in data collection. These include the origin of MSCs (human or mouse), tissue source, route of administration, timing of treatment, number of repetitions, dosage, and mouse strains, which are all critical and have different effects on the therapeutic outcome.

The in vivo studies related to the effects of MSCs and MSC-Exo on the cells relating to innate and adaptive immunity, described in this section, are summarized in Table 3.

## 8. Clinical Studies

Several clinical studies have demonstrated the promising therapeutic effects of various types of MSCs in psoriasis treatment. Chen et al. used UC-MSCs to treat two patients with psoriasis vulgaris in prolonged remission. One patient received a single dose of UC-MSCs following autologous hematopoietic stem cell transplantation, resulting in complete skin recovery within 12 months and remaining without recurrences for 5 years post treatment. The second patient was administered three UC-MSC infusions over three weeks, followed by two additional infusions as a consolidation therapy, and remained symptom-free for four years [144]. Similar effects were observed in a study where a patient with severe plaque psoriasis received five weekly intravenous infusions of gingival-tissue-derived MSCs (GMSCs). They experienced a complete regression of psoriatic lesions and stable remission for up to three years [145].

Some clinical studies employed AD-MSCs. For instance, De Jesus et al. investigated the use of autologous AD-MSCs in two patients, one with psoriasis vulgaris (PV) and the other with psoriatic arthritis (PA). The authors reported substantial improvements in both patients, as evidenced by decreased PASI scores and prolonged maintenance post treatment [146]. A recent study illustrated that two patients with moderate to severe psoriasis treated with monthly intravenous AD-MSC injections for three months achieved significant improvements. Their PASI scores reduced by half and the therapy was safe and well-tolerated during the yearly treatment [87]. Additionally, five patients experiencing psoriatic plaques underwent treatment with subcutaneous injections of allogeneic AD-MSCs. They all had considerable progress in skin thickness, redness, and scaling. The authors reported other improvements, such as reduced inflammatory markers in the dermis, demonstrating the modulation of inflammation and lack of adverse effects [147].

Other clinical studies have explored different sources and methods of MSC therapy for psoriasis. In one study, autologous stromal vascular fraction (SVF), which contains AD-MSCs, was used to treat a patient with severe psoriasis. During a year, his PASI score dropped dramatically from 50.4 to 0.3, with significant improvements in skin quality [148]. Likewise, mesenchymal-stem-cell-conditioned medium (MSC-CM) was applied topically in a patient suffering from scalp psoriasis. After a month-long application, the patient’s Psoriasis Scalp Severity Index (PSSI) score decreased from 28 to 0. Their disease-free state lasted for six months, showcasing the potential for topical MSC-based therapies [149].

The immunomodulatory effects of UC-MSCs have also been pointed out in several studies. Cheng et al. conducted a phase 1/2a clinical trial in 2022 where 17 patients with psoriasis received a UC-MSC infusion, resulting in a 47.1% PASI improvement in some, including 35.3% who experienced over a 75% PASI improvement. This study also further showed immunomodulation of the immune responses, characterized by increases in Tregs and memory T cells while reducing Th17 cells and serum IL-17 levels [150]. Similarly, Güler et al. performed a treatment on a patient who suffered from psoriasis and aplastic anemia using UC-MSCs in tandem with allogenic bone marrow transplantation for graft-versus-host disease prophylaxis; this resulted in the complete remission of psoriasis, and no recurrence was seen in the 150 days post-transplant [151].

There have also been clinical studies that have evaluated the safety and efficacy of adipose-tissue-derived MSC-Exo for the treatment of psoriasis. In one phase I/II clinical study, patients were administered exosomes intradermally at three dosages (50 µg, 100 µg, and 200 µg per cm^2^ of psoriatic skin), and the results were examined using the Target Lesion Assessment Score (TLAS), histopathological analysis, and immunohistochemistry. The study showed that the 200 µg dose substantially reduced erythema, induration, and lesion thickness. Additionally, while the inflammatory markers such as IL17, TNF-α, and CD3 decreased, anti-inflammatory marker FOXP3 increased [152].

Treatment of psoriasis using MSCs, including MSC-Exo, has shown significant promise in effectively reducing psoriasis symptoms and maintaining long-term remission with minimal side effects. It is proposed that further research should focus on large-scale trials and optimization of treatment protocols to fully integrate MSCs and MSC-Exo into standard psoriasis treatments. Data from clinical studies on the therapeutic properties of MSCs and MSC-Exo in the treatment of patients with psoriasis, described in this section, are summarized in Table 4.

## 9. Conclusions

Psoriasis is a chronic inflammatory autoimmune skin disease that develops as a result of inadequate activation of the cellular component of the immune system. The activation of a subpopulation of T-lymphocytes is a key event in the pathogenesis of psoriasis. Disruption of the processes of their proliferation and differentiation is considered a consequence of the excessive production of cytokines, chemokines, and growth factors that enhance skin lesions. Modern treatment approaches do not solve the cause of the disease or relieve symptoms. The anti-inflammatory and immunomodulatory properties of MSCs have been successfully tested for the treatment of autoimmune diseases in preclinical and clinical studies. Normalizing the immune status can reveal the cause of psoriasis and is the most promising treatment method. The use of MSC-Exo is a more convenient and safe treatment method and is currently being tested in a number of clinical studies. The therapeutic effect of MSCs can be heterogeneous depending on delivery protocols, MSC populations, and cell drug delivery conditions. MSCs are capable of inducing various immunomodulatory responses that depend on the proinflammatory or immunosuppressive phenotypes. Other limitations include their ability to spread, lifespan, potential for contamination, and rejection ability. Recently, studies have confirmed that MSCs achieve a therapeutic effect in vivo mainly due to paracrine signaling through the secretome (through exosomes), rather than regenerative abilities. This has increased the priority given to using exosomes rather than MSCs.

## Figures and Tables

**Figure 1 biomolecules-14-01351-f001:**
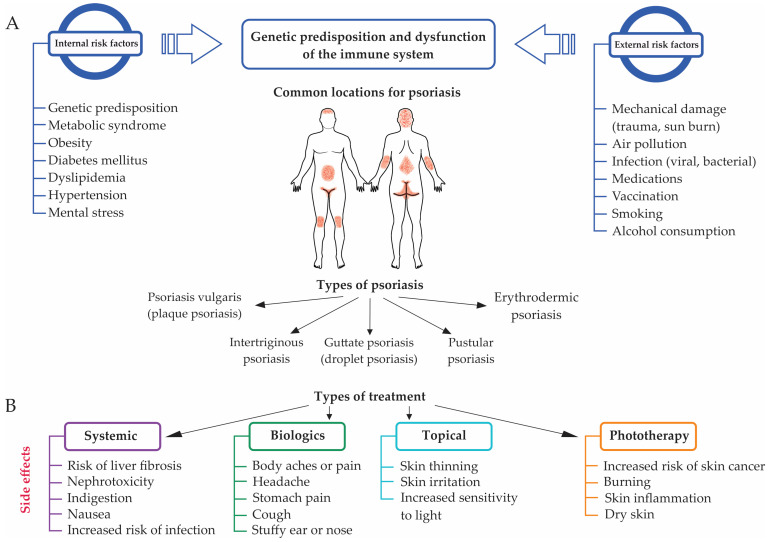
General information about psoriasis: (**A**) Internal and external risk factors for psoriasis, common locations on the body, and types of psoriasis. (**B**) Types of treatments and their side effects.

**Figure 2 biomolecules-14-01351-f002:**
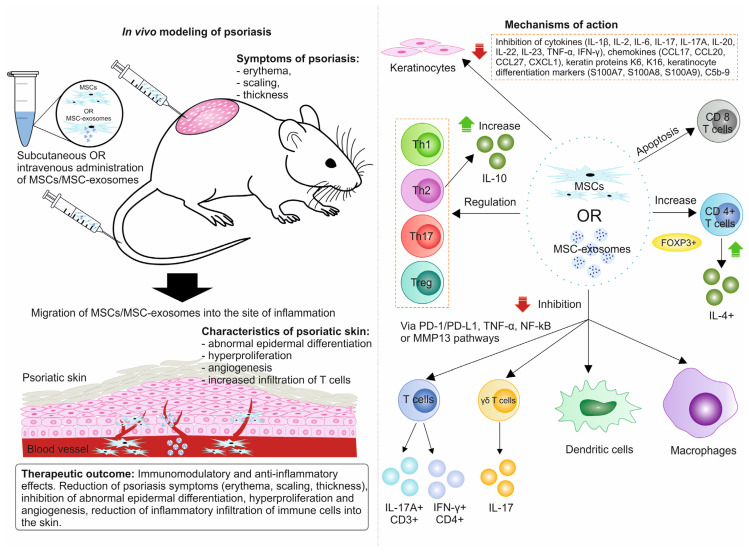
Mechanisms of action of MSCs and/or MSC-Exo in animal models of psoriasis-like skin inflammation.

**Table 1 biomolecules-14-01351-t001:** Treatment methods for psoriasis.

Treatment Method		Therapeutic Effect	Reference
Biologics	IL-17, TNF-α, IL-12, IL-23 inhibitors (e.g., infliximab, ixekizumab, secukinumab, apremilast)	Significant improvement in psoriasis severity (PASI 100 response) and long-term control of symptoms, including sustained inhibition of radiographic progression in psoriatic arthritis patients.	[79,80,82,91]
	IL-17 + TNF-α combination therapy	Enhanced reduction in skin lesions and better control of systemic inflammation compared to monotherapy with either inhibitor.	[79]
Systemic therapy	Corticosteroid injections	Directly reduces inflammation and controls severe flare-ups, but may cause localized side effects such as atrophy.	[92]
	Oral therapy: acitretin, cyclosporine, methotrexate	Reduces general symptoms of psoriasis, can replace other methods that were restricted.	[81]
Topical therapy	Topical corticosteroids	Significant reduction in inflammation, plaque thickness, and erythema. Long-term use may lead to skin thinning and other side effects.	[93]
	Ointments, vitamin D analogues (e.g., calcipotriene, calcitriol)	Slows down skin cell proliferation, reduces plaque formation, and is often used in combination with corticosteroids for enhanced effect.	[84]
Photo-therapy	Narrowband ultraviolet B (NB-UVB)	Reduces inflammation and slows the rapid growth of skin cells. Effective in treating moderate to severe plaque psoriasis with fewer side effects compared to broadband UVB.	[85]
	Psoralen plus UVA (PUVA)	Effective for treating severe psoriasis, especially in cases resistant to other treatments. PUVA slows skin cell turnover and reduces scaling and inflammation.	[94]
	Excimer laser (Targeted UVB)	Specific targeting of psoriasis plaques with minimal off-target effects. Particularly useful for localized psoriasis, including scalp and nails.	[95]
Cell therapy	MSCs	MSCs have been shown to reduce inflammation and modulate immune responses in psoriasis, leading to improved skin healing, reduced symptoms, and increased damaged tissue regeneration.	[88]
	Adipose-derived MSCs (AD-MSCs)	AD-MSCs exhibit immunomodulatory properties, suppressing inflammatory cytokines such as IL-17 and TNF-α, leading to the improvement of psoriatic lesions and a reduction in disease severity.	[87]
	Human umbilical cord MSCs (hUC-MSCs)	hUC-MSCs have shown efficacy in reducing inflammation and promoting tissue repair, specifically by reducing IL-17-producing γδ T cells and modulating the immune response to alleviate psoriasis symptoms.	[86]
	MSC-Exo	MSC-Exo attenuates inflammation and promotes tissue regeneration by modulating immune responses through decreasing proinflammatory cytokines like TNF-α and IL-17.	[89]
	Adipose-derived stem cell exosomes (ASC-Exo)	ASC-Exo reduces psoriatic lesions and improves skin barrier function. ASC-Exo have shown efficacy in reducing hyperpigmentation and improving overall skin health.	[90]
Other	Changing diet manner	Choosing a specific form of balanced diet (plant-based) could help to reduce overweight in patients with psoriasis and their severity.	[79]

**Table 2 biomolecules-14-01351-t002:** Therapeutic and immunomodulatory effects of cytokines, growth factors, enzymes, and other molecules produced by MSCs.

Cytokine	Type of MSC	Preclinical Model	Therapeutic and Immunomodulatory Effects	Reference
IFN-γ	Bone marrow MSCs (BM-MSCs)	Solid organ transplantation models	IFN-γ primes MSCs to enhance immunosuppressive properties by upregulating IDO and iNOS expression, leading to suppression of T-cell proliferation.	[107]
	Human MSCs (hMSCs)	Graft-versus-host disease (GVHD) models	Enhances MSC-mediated immunosuppression and improves outcomes in GVHD by increasing programmed death ligand 1 (PD-L1) expression.	[108]
TNF-α	hMSCs	Myocardial infarction in rats	Enhances the therapeutic efficacy of MSCs by promoting NO production and reducing apoptosis, thereby improving heart function.	[109]
IL-1β	hMSCs	Inflammatory models in vitro	IL-1β primes MSCs to secrete anti-inflammatory mediators like PGE2, enhancing their immunomodulatory functions.	[110]
IL-2	hMSCs	Cancer immunotherapy models	IL-2 priming enhances MSCs’ ability to inhibit NK cell proliferation and promote Tregs’ differentiation, leading to improved immunosuppression.	[111]
IL-17 + TNF-α	BM-MSCs	Autoimmune encephalomyelitis in mice	The combination of IL-17 and TNF-α significantly enhances the immunosuppressive effects of MSCs, leading to reduced inflammation and better control of autoimmune responses.	[112]
TGF-β	hMSCs	Cecal ligation and puncture (CLP)-induced sepsis in mice	Attenuation of histopathological damage to the organ, reduction in the level of proinflammatory cytokines, and inhibition of macrophage infiltration into tissues.	[113]
	BM-MSCs	Bone metabolism and tissue regeneration	Promotes osteoblast differentiation, bone remodeling, and tissue regeneration.	[114]
HGF	BM-MSCs	Spinal cord injury in rats	Promotes recovery by reducing inflammation and enhancing nerve regeneration.	[115]
	Gene-modified MSCs	Ischemia/reperfusion-induced acute lung injury in rats	Reduction in lung injury, and improvement of cell survival and tissue repair through anti-apoptotic effect.	[116]
PGE2	BM-MSCs	LPS-induced acute lung injury (ALI) in mice	Enhancement of the protective effects of MSCs by modulating macrophage polarization, reducing inflammation, and improving lung function.	[117]
	Human-embryonic-stem-cell- derived MSCs	Liver fibrosis in mouse models	Reduces liver fibrosis through immunosuppressive mechanisms and regulation of T-cell function.	[118]
IDO	BM-MSCs	Heart allograft rejection in rats	Suppresses heart allograft rejection by increasing the production and activity of DCs and Tregs.	[119]
	hUC-MSCs	Dilated cardiomyopathy (DCM) in rats	Enhances cardiac function and survival by upregulating IDO expression, which modulates immune response and reduces inflammation.	[120]
iNOS	BM-MSCs	Systemic sclerosis (SSc) in mice	iNOS activity is crucial for the anti-fibrotic effects of MSCs, reducing skin thickness and collagen deposition.	[121]
	hMSCs	Myocardial infarction in rats	Enhances survival and paracrine function of MSCs, leading to reduced apoptosis and improved heart function.	[109]

**Table 3 biomolecules-14-01351-t003:** In vivo studies of MSCs and MSC-Exo for the therapy of psoriasis.

**Psoriasis Model**	**Animals, Age**	**Source and Tissue Origin** **of MSCs**	**Route of** **Administration/** **Number of** **Repetitions**	**Dosage**	**Mechanism of** **Action**	**Therapeutic Outcome**	**Reference**
IMQ-induced psoriasis-like skin inflammation	C57BL/6 mice, 8 weeks	Human-umbilical-cord-blood- derived MSCs (hUCB- MSC)	Subcutaneous injection, 2 times (24 h before and at day 6 of IMQ application)	2 × 10^6^ cells	MSCs overexpress SOD3, to prevent the severity and progression of psoriasis through the regulation of immune cell infiltration and functions, specifically DCs, neutrophils, and Th17 cells, and by regulating epidermal functions, TLR-7-dependent and independent pathways, MAP kinases, and JAK-STAT pathways, which augment the inflammatory actions.	Immunomodulatory effect, reducing the thickness of the epidermis and inhibiting the infiltrations of various immune cells into the skin, spleen, and lymph nodes.	[129]
IMQ-induced psoriasis-like skin inflammation IL-23-mediated psoriasis-like skin inflammation	C57/BL mice, male, 8–12 weeks	hUCB- MSCs	Subcutaneous injection, 24 h before IMQ application Subcutaneous injection, near the ear—2 times (24 h before and day 7) Subcutaneous injection (back skin)—2 times (on days 7 and 13)	2 × 10^6^ cells (per injection)	hUCB-MSCs inhibit proinflammatory cytokine and chemokine gene expression and suppress Th17 cell differentiation. MSCs prevent the infiltration of immune cells (CD4+ T cells, CD11b+ cells, and CD11c+ cells) into the skin and inhibit the expression of proinflammatory (IL-1β, IL-6, IL-17, IL-22, IL-20, TNF-α) and chemokine genes in the skin.	Anti- inflammatory effect and regulatory effect on immune cells.	[134]
IMQ-induced psoriasis-like skin inflammation	C57BL/6 mice, female, 8 weeks	T-MSC	Intravenous administration, 2 times (on days 1 and 3 of the IMQ application period)	1 × 10^6^ cells	T-MSCs induce a blockade of PD-L1, which leads to downregulation in gene expression of IL-23, TNF-α, IFN-γ, IL-17, IL-22, K6, K16, and CCL20.	Immunosuppressive effect.	[135]
IMQ-induced psoriasis-like skin inflammation	BALB/c mice, female, 8 weeks	hUC- MSCs	Intravenous administration after 6 consecutive days of IMQ application	1 × 10^6^ cells	MSCs infusion inhibited the infiltration of immune cells, T cells (CD3^+^ cells), neutrophils (Gr-1^+^ cells), and IL-17^+^ cells into the skin. MSCs reduce the expression level of proinflammatory cytokines (IL-17, IL-23, IL-6, and IL-1β) and keratinocyte differentiation markers (S100A7, S100A8, and S100A9) and also increase the expression level of anti- inflammatory cytokine IL-10.	Immunomodulatory and anti-inflammatory effects.	[136]
IMQ-induced psoriasis like skin inflammation	C57BL/6J mice	hAMSC	Injection into each mouse ear	5 × 10^5^ cells	hAMSC alleviates the keratinocyte response to proinflammatory cytokines IL-17A and IL-22 and chemokine CXCL1.	Immunomodulatory effect.	[137]
IMQ-induced psoriasis-like skin inflammation	BALB/c mice, male, 6–9 weeks	MSC-Exo	Daily topical application on day 3 and for a total of 3 or 7 days (two experiments using different batches of exosomes)	100 μg/mL, 200 μL per mouse	Topically applied MSC-Exo reduce the proinflammatory cytokines (IL-17, IL-23) and inhibit C5b-9 activation through CD59 in the stratum corneum.	Anti-inflammatory effect.	[138]
IMQ-induced psoriasis-like skin inflammation	C57BL/6 mice, 8 weeks	hUC-MSC-derived exosomes (hUC-MSC-Exo)	Subcutaneous injection	50 μg per mouse	hUC-MSCs-Exo suppress the expression of IL-17, IL-23, and CCL20, thereby inhibiting the phosphorylation of STAT3.	Immunomodulatory effect. Reduction in psoriatic erythema, scaling, thickening, inflammatory infiltration, and inhibition of epidermal hyperplasia.	[131]
IMQ-induced psoriasis-like skin inflammation	-	ASCs	Intradermal injections into the dorsal areas, 3 times (on days 0, 3, and 4)	1 × 10^6^ cells	ASCs inhibit the production of Th17-associated cytokines, such as IL-17A and TNF-a.	Immunosuppressive effect.	[139]
IMQ-induced psoriasis-like skin inflammation	C57BL/6 mice, female, 6–8 weeks	MSC-Exo-PD-L1	Intravenous administration, 4 days	50 μg per mouse	MSC-Exo-PD-L1 restore tissue lesions by inhibition the inflammatory immune cells via the PD-1/PD-L1 pathway.	Anti-inflammatory effect and regulatory effect on immune cells.	[140]
IMQ-induced psoriasis-like skin inflammation	C57BL/6 mice, 6–8 weeks	hUC-MSCs	Intravenous (one day before the first IMQ application) or subcutaneous (three times: 1 day before, 2 days after, and 4 days after the first application of IMQ) administration	5 doses (0.5; 1; 2; 5 or 10 × 10^6^ cells)	hUC-MSCs suppress skin inflammation by inhibiting γδ T cells producing IL-17.	Significant reduction in the severity of psoriasis-like dermatitis and suppression of the inflammatory response of cells.	[86]
IMQ-induced psoriasis-like skin inflammation	C57Bl/6 J mice, female, 7–9 weeks	hUC-MSCs	Intravenous administration, 5 days	4 × 10^5^ cells in 200 μL PBS	hUC-MSCs inhibit TNF-α production by monocytes and macrophages, which in turn prevents keratinocyte proliferation induced by the TNF-α/NF-κB/MMP13 axis.	Significant reduction in epidermal thickening and excessive keratinocyte proliferation.	[141]
IMQ-induced psoriasis-like skin inflammation	C57BL/6 mice, female, 7 weeks	BM-MSCs or AD-MSCs	Intravenous administration after 4 days of IMQ application	1 × 10^6^ cells/mouse in 100 μL of PBS	BM- or AD-MSCs improve skin repair by increasing the levels of IL-17A and TGF-β, and TGF-β promotes controlled differentiation of keratinocytes.	Accelerated healing process and reduction in severity of psoriasis.	[142]
IMQ-induced psoriasis-like skin inflammation	C57BL/6 mice, 8–12 weeks	UCB-MNC-Exo	Topical application, one hour after each imiquimod application for 6 days	3 × 10^9^ particles/cm^2^ UCB-MNC-sEV dissolved in hydrogel	UCB-MNC-Exo increases the number of Tregs in the skin.	Reduction in acanthosis, and prevention of keratinocyte hyperproliferation.	[132]
IMQ-induced psoriasis-like skin inflammation	Albino rats, male	hUC-MSCs	Subcutaneous injections at the four corners around the edge of the inflamed area of the skin, after 6 days of IMQ application	2 × 10^6^ cells per injection within 2 mL of the media	MSCs regulate immune cell infiltration (Th17 cells), epidermal functions, and differentiation.	Immunomodulatory and anti-inflammatory effects.	[143]

**Table 4 biomolecules-14-01351-t004:** Clinical studies of MSCs and MSC-Exo for the treatment of psoriasis.

**Disease**	**Source of MSCs**	**N** **(M, F)**	**Age ^1^** **(M, F)**	**Dosage**	**Adverse** **Effects**	**Results**	**Reference**
Psoriasis vulgaris	Allogeneic hUC-MSCs	1M, 1F	35, 26	2–3 IV infusions at a dose of 1 × 10^6^ cells/kg	Not observed	Remained relapse-free for 4–5 years	[144]
Plaque psoriasis	Allogeneic hGMSCs	1M	19	5 IV infusions at a dose of 3 × 10^6^ cells/kg	Not observed	Remained disease-free for 3 years	[145]
Psoriasis vulgaris and arthritis	Autologous hAD-MSCs	1M, 1F	58, 28	PV—3 IV infusions (2.36 × 10^6^ cells/kg) PA—2 IV infusions (5.3 × 10^5^ cells/kg)	Not observed	PV—decreased PASI for 292 days PA—remained relapse-free for 2 years	[146]
Psoriasis vulgaris	Allogeneic hAD-MSCs	6M, 1F	50.71 ± 10.45	3 IV infusions at a dose of 0.5 × 10^6^ cells/kg	Mild or moderate AEs	PASI-50 in two patients with no additional treatment	[87]
Plaque psoriasis	Allogeneic hAD-MSCs	3M, 2F	32.8 ± 8.18	Subcutaneous injection at a dose of 1–3 × 10^6^ cells/cm^2^	Mild burning and pain in some patients	Decreased PASI after 6 months	[147]
Severe psoriasis	Autologous SVF, hAD-MSCs	1M	43	IV infusion of around 30–60 × 10^6^ cells	Not observed	PASI dropped from 50.4 to 0.3 in a month, sustained for a year	[148]
Psoriasis vulgaris	MSC-CM	1M	38	Daily topical application for one month	Not observed	PSSI dropped from 28 to 0 in a month, sustained for 6 months	[149]
Psoriasis vulgaris	Allogeneic hUC-MSCs	8M, 9F	40.76 ± 8.85	4 IV infusions at a dose between 1.5 and 3.0 × 10^6^ cells/kg	Mild adverse effects all resolved within a day	PASI improvements in all patients; one patient with PASI-50 for almost 3 years	[150]
Psoriasis	UC-MSCs	1M	12	N/A	No complications during the follow-up	Complete remission of psoriasis lesions by day 7	[151]
Plaque psoriasis	hAD- MSC-Exo	7M, 3F	36.6 ± 8.07	200 µg MSC-Exo	No adverse effects observed	Reduced erythema, induration, and lesion thickness	[152]

^1^ Mean ± standard deviation. N/A—not applicable.

## Data Availability

The data presented in this study are available on request from the corresponding author.

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
