# Peer review of "Psoriasis: The Versatility of Mesenchymal Stem Cell and Exosome Therapies"

_biomolecules, 2024, doi:10.3390/biom14111351_

Round 1

Reviewer 1 Report

Comments and Suggestions for Authors

General consideration. There is a great deal of bibliographic research in the article but personally I would have preferred less bibliographic data but a more extensive characterization of both the MSCs and the molecular and cellular mechanisms induced by these cells in modulating the immune response in psoriasis to provide therapeutic benefits

Line 36 Initially I suggest introducing a phase like this: the etiopathology of psoriasis is not yet clear but it seems that genetic, immunological and environmental factors are involved.

 Line 62: add possible limitation about use of MSCs, if there are, as : low cells survival and engraftment rates after the transplant process, scarcity in tissues, mutations, heterogenic nature in cell source, and long terms safety of therapeutic advantages to be verified

Line 86: I suggest expanding the chapter by talking about Psoriasis: the involvement of the IL-23/IL-17 axis; the role of IL-23, secreted by dendritic cells and monocytes/macrophages, in promoting the expansion of IL-17-producing cells. The involvement of TLRs is also important as such as the induction of autoreactive T lymphocytes in the disease against autoantigens such as LL37 (complexes of LL37 with nucleic acids active TRLs). The neutrophils participate (by Nets) in the exacerbation of diseases as the main cellular source of IL-17 and autoantigens (LL37) during NETosis. The major inducer of NETosis is the activated complement. Important is also the imbalance between T cells/Tregs and INF signature of disease.

General consideration: Personally, I would have spent more time talking about the hypothesized causes of the disease than about the diagnosis. An introduction about psoriasis can help the reader to understand better the article and mechanisms of therapeutic action of MSCs

Line 168 and table 1: please amplify treatments for psoriasis for a complete information, if possible. They are devised in topical, systemic and biological (Christopher E M Griffiths, April W Armstrong, Johann E Gudjonsson, Jonathan N W N Barker,Psoriasis. The Lancet,Volume 397, Issue 10281,  2021,Pages 1301-1315, ISSN 0140-6736, https://doi.org/10.1016/S0140-6736(20)32549-6.)

Line 182: Fig 1 lists some risk factors of psoriasis, but not all. The main cause of the disease is not mentioned either in the introduction or in the figure: a dysfunctional immune system and impaired functionality of TRLs. Psoriasis is at first an autoimmune disease. Add between internal risk factors a dysfunctional immune system.

Line 187.  The chapter is not clear; if possible, rewrite it, or add a phase in which you underlie that MSCs have a  basic properties in modulating the immune system and the additional ones induced by an inflamed environment (Jiang W, Xu J. Immune modulation by mesenchymal stem cells. Cell Prolif. 2020 Jan;53(1):e12712. doi: 10.1111/cpr.12712. Epub 2019 Nov 15. PMID: 31730279; PMCID: PMC6985662). In this chapter, I would have listed and explain the major mechanisms attributed to MScs in inducing immunosuppressive effects in Psoriasis or in general. As such as, for a complete information, add that although MSCs work essentially by extracellular vesicles (EVs), it is also important to say that these cells may also affect immune cells by directly interaction with them. For example, ICAM-1 expressed by MSCs, is able to interact with LFA-1 on T cells and NK cells and reduce their ability to attack other cells. Inflammatory microenvironment may promote up regulation of these molecules on MSCs. Add this information to have a complete vision of the argument.  1) Ren, G. et al. Inflammatory cytokine-induced intercellular adhesion molecule-1 and vascular cell adhesion molecule-1 in mesenchymal stem cells are critical for immunosuppression. J. Immunol. 184, 2321–2328 (2010);  2) Kyurkchiev D, Bochev I, Ivanova-Todorova E, Mourdjeva M, Oreshkova T, Belemezova K, Kyurkchiev S. Secretion of immunoregulatory cytokines by mesenchymal stem cells. World J Stem Cells. 2014; 6(5):552–70. 56).

Line 194. Add also neutrophils.

Line 204: “Therapeutic and immunomodulatory effects of cytokines, growth factors, enzymes and other molecules produced by MSCs.” Are you sure that MSCs produce all these molecules ? Are they molecules present in an inflamed environment? Are molecules produced in part by MSCs and in part by environment? Please correct and clarify

266:  Add also other information and literature: MSCs from the skin of individuals with psoriasis (pD-MSCs) showed a higher expression of genes responsive for encoding Th1 and Th17  cytockines than normal dermal MSCs nD-MSCs) . (Imbalance of Th21-Th17 /Th2 in pD - MSCs.). Moreover,  nD-MSCs have a more potent inhibitory effect on T cell proliferation than pD- MCSs (Zhao et al. Imminomodulation effect of psoriasis derivated dermal mesenkimal stem cells on Th1/Th17 cell.  Eur. J Dermatol. 31 (3) 2021 : 318- 325.). In particular nD-MSCs have greter efficacy compared to pD-MSCs in suppressing the proliferation of CD3 T cells and inducing their apoptosis. These cells can also reduce the Th1/Treg ratio and improve the immunosuppressive function of Tregs.  nD-MCSs can also outperformance pD-MSCs in increasing TGF-b receptor partially restoring the biological immunosuppressive function of psoriatic CD3T cells throught upregulation of TGF-b receptor/signaliing . (J. Jiao, et al., Normal mesenchymal stem cells can improve the abnormal function of T cells in psoriasis via upregulating transforming growth factor-β receptor,J. Dermatol. 49 (10) (2022) 988–997.

Line 355: Table 3: Please add the bibliography and the mechanism of action sequentially in the table with the same sequence exposed in the “In vitro studies chapter”. Bibliography number 85, 86 72 is missing

Line 418: Table 4. The references are not reported sequentially as in the chapter "clinical studies" and some are missing 90. Delete the 98.

Line 430: if you mention that exosome use is more convenient and safe, briefly explain why.

Author Response

 Comments for Reviewer 1

Dear Reviewer, thank you for your useful comments and remarks, which will help us improve our work. We have tried to correct all the remarks. In the table you can find our comments and the location of the corrections in the manuscript of the article. Thank you very much for your work!

No.

Reviewer's comment

Reply to comment

Status

1.

Line 36: Initially I suggest introducing a phase like this: the etiopathology of psoriasis is not yet clear but it seems that genetic, immunological and environmental factors are involved.

Part of the first paragraph has been rewritten. New data has been added, and your suggestion has been taken into account.

«Psoriasis is one of the most common clinically heterogeneous immune-mediated inflammatory lifelong skin diseases [1,2]. Psoriasis is characterized by hyperproliferative keratinocytes and infiltration of T cells, dendritic cells (DCs), macrophages and neutrophils [1]. The etiopathology of psoriasis is not yet clear but it seems that genetic, immunological and environmental factors are involved [3,4].».

Corrected

2.

Line 62: add possible limitation about use of MSCs, if there are, as: low cells survival and engraftment rates after the transplant process, scarcity in tissues, mutations, heterogenic nature in cell source, and long terms safety of therapeutic advantages to be verified.

Your comments have been taken into account, and as a result, information about possible limitations associated with the use of MSCs has been added.

«In addition to numerous advantages and benefits, there are a number of problems limiting the widespread use of MSC therapy in clinical practice, which include: risk of developing tumors, transmission of viruses and prions after stem cell transplantation; long-term in vitro cultivation of MSCs leads to loss of the potential of MSCs for differentiation and morphological changes, and also increases probability of malignant transformation; low survival and engraftment rates of MSCs due to their short-lived viability after injection; low therapeutic effect and increased immunogenicity in differentiated MSCs; heterogeneity of MSCs is due to differences in the health status, genetics, gender and age of donors; varying degrees of stem cell stability and differentiation capacity between MSCs isolated from different sources; different levels of expansion ability under different culture conditions; immune compatibility between donors and recipients; ethical issues; high manufacturing costs [12-15]. Consequently, long-term research and monitoring will be required to examine the long-term effects of MSC therapy, including any negative effects [12].».

Additionally: Additional information has been added regarding the benefits of MSC-exosomes compared to cell therapy.

«Along with MSCs, the therapeutic effects of their exosomes (MSC-Exo) are being extensively investigated. This is mostly because, stem cell-derived exosomes have various advantages over stem cells, including non-immunogenicity, non-infusion toxicity, easy access, easy storage, and the absence of tumorigenic potential and ethical problems [16]».

MSC-Exo are used to mediate tissue regeneration in a variety of diseases, such as cutaneous wound healing [19 Hu, 2022], cancer [20-22], ischemic heart disease, lung diseases [23], liver fibrosis, neurological, autoimmune, and inflammatory diseases [18]. As already mentioned, a significant advantage of MSC-Exo compared to MSCs is that they overcome the limitations of cell therapy by providing comparable benefits in a safer and more stable extracellular vesicle format. Moreover, the contents of exosomes can also be modified to enhance regenerative biological activity [24].».

Corrected

3.

Line 86: I suggest expanding the chapter by talking about Psoriasis: the involvement of the IL-23/IL-17 axis; the role of IL-23, secreted by dendritic cells and monocytes/macrophages, in promoting the expansion of IL-17-producing cells. The involvement of TLRs is also important as such as the induction of autoreactive T lymphocytes in the disease against autoantigens such as LL37 (complexes of LL37 with nucleic acids active TRLs). The neutrophils participate (by Nets) in the exacerbation of diseases as the main cellular source of IL-17 and autoantigens (LL37) during NETosis. The major inducer of NETosis is the activated complement. Important is also the imbalance between T cells/Tregs and INF signature of disease.

To address this remark, a chapter on the mechanism of disease occurrence was added after the introduction chapter.

«2. Disease pathogenesis mechanism

Psoriasis pathogenesis is governed by the interleukin (IL)-23/IL-17 signaling axis [25]. This complex signaling mechanism involves members of both the innate and adaptive immune systems. Disease onset begins with immune activation in genetically pre-disposed individuals following environmental triggers such as infection, medication, and smoking [26]. Another concomitant event triggering disease initiation is the loss of immune tolerance through the recognition of autoantigens, specifically antimicrobial pep-tides like LL-37/cathelicidin released by keratinocytes [27]. Individuals with genetic susceptibility also release self-nucleotides that can form complexes with LL-37 and are recognized by toll-like receptors (TLR7 and TLR9) on the surface of plasmacytoid dendritic cells (pDCs) [28,29]. This binding event activates pDCs, eliciting the secretion of inflammatory mediators interferon-α (IFN-α) and IFN-β to stimulate other dermal DC subsets to produce proinflammatory mediators, such as the primary cytokine IL-23, IL-12 and tumor necrosis factor (TNF) [30]. Activated pDCs and other DC subsets present the psoriatic autoantigen LL-37/cathelicidin to CD4⁺ and CD8⁺ T cells. Antigen presentation can occur within the dermis, stimulating resident memory T cells, and in the draining lymph nodes, where it activates naïve T cells. At the same time, secreted IL-23 evokes the further activation and clonal expansion of IL-17- and IL-22-secreting T helper (Th)17 and Th22 cells, respectively [31].

Active Th17 cells exert their downstream effect through several cytokines IL-17, IL-26, IL-29 and TNF-α. They play a significant role in creating a feed-forward loop that exacerbates the disease state by recruiting other cell types. First, their key cytokine IL-17 targets keratinocytes that express IL-17 receptors, inducing the expression of CC-chemokine ligand 20 (CCL20) [32]. This chemokine attracts IL-23-producing DCs and Th17 cells, further compounding the already inflamed environment. IL-17 drives disease pathogenesis by activating psoriasis-related genes in keratinocytes via IL-22, IL- 19, and IL-36 to increase epidermal hyperplasia and produce more antimicrobial peptides, including LL-37/cathelicidin [33]. IL-17 also promotes and maintains an inflammatory environment by attracting additional innate immune cell populations. In particular, circulating neutrophils aggregate at the inflamed site due to the release of neutrophil-attracting factors such as chemokine (C-X-C motif) ligand (CXCL)1/2/3/5 and CXCL8 [34].

While the IL-23/IL-17 signaling axis is pivotal in disease onset and progression, re-cent studies have explored the active involvement of neutrophils in psoriasis pathogenesis. Neutrophils employ a distinct mechanism called NETosis to eliminate foreign bodies. NETosis is a form of cell death in which neutrophil extracellular traps (NETs) - web-like structures composed of cytosolic proteins and decondensed DNA/RNA - are released into the surrounding environment [35,36]. Upon receiving inflammatory stimuli, recruited neutrophils actively form NETs within psoriatic lesions. Notably, the severity of psoriasis correlates with the quantity of NETs in blood samples [37 Hu S., 2016]. These NETs are abundant in LL-37 and RNA. LL-37 can form complexes with RNA, facilitating RNA's uptake by neutrophils. This process subsequently activates TLRs and leads to the secretion of IL-8, a neutrophil chemotactic factor that recruits additional neutrophils to the lesion site [38]. Intriguingly, the same complex can induce neutrophils to release more NETs, thereby propagating the inflammatory cycle [39].

Thus, in the IL-23/IL-17 disease model, dermal DCs release IL-23, which eventually induces Th17 cell activation and proliferation. Th17 cells produce pro-inflammatory cytokines to target keratinocytes that sustains and enhances the chronic inflammatory state by generating additional IL-23, as well as other pro-inflammatory cytokines, chemokines, S100 family proteins, and antimicrobial peptides. This repetitive cycle perseveres and amplifies the ongoing inflammatory psoriatic process.».

Corrected

4.

Line 168 and table 1: please amplify treatments for psoriasis for a complete information, if possible. They are devised in topical, systemic and biological (Christopher E M Griffiths, April W Armstrong, Johann E Gudjonsson, Jonathan N W N Barker,Psoriasis. The Lancet,Volume 397, Issue 10281,  2021,Pages 1301-1315, ISSN 0140-6736, https://doi.org/10.1016/S0140-6736(20)32549-6).

We accepted your wish and expanded the methods of treating psoriasis, including information on cell therapy for psoriasis. The table 1 has also been significantly revised. The literature you recommended has been studied and added.

«The decision on the treatment methods should be taken according to standard guidelines, tended and personalized considering the disease history of patient [46]. Depending on the severity of the disease, treatment ways includes systemic therapy that were presented by biologics, namely with monoclonal antibodies, which are IL-17, tumor necrosis factor alpha (TNF-α), IL-12, IL-23 inhibitors (e.g., infliximab, ixekizumab, secukinumab, apremilast) or corticosteroid injections; oral therapy by immunosuppressants (methotrexate, cyclosporine A, etc.); topical therapy by surface preparations (creams containing steroid hormones, vitamin D3, etc.); phototherapy (irradiation with ultraviolet light), and selective toxins to lymphocytes (DAB389IL-2) [46,79-85 Griffiths C.E.M., 2021]. Also, cell therapy could be an effective and safe way in treating the disease. In particular, the application of MSCs themselves, and MSC de-rived exosomes play a crucial role in modulating inflammation and tissue repair in psoriasis [86-90].».

Corrected

5.

Line 182: Fig 1 lists some risk factors of psoriasis, but not all. The main cause of the disease is not mentioned either in the introduction or in the figure: a dysfunctional immune system and impaired functionality of TRLs. Psoriasis is at first an autoimmune disease. Add between internal risk factors a dysfunctional immune system.

Your comments on improving Figure 1 have been accepted and revised, and the structure of the figure has been significantly changed compared to the original version.

Additionally: We also tried to describe in detail the role of the genetic factor, as well as internal factors, in the development of this disease. We decided to include this section in Chapter 3.

«Genetic factors play a substantial role in the onset and progression of psoriasis. Certain populations have genetic predispositions that make them more susceptible to developing it. Previous genome-wide studies identified several major susceptibility loci. Among them, the major histocompatibility complex (MHC) region, particularly class I HLA genes like HLA-C, is strongly associated with psoriasis. The HLA-C*06:02 allele is especially significant, being identified as a primary genetic factor in psoriasis susceptibility across different populations [47].

Genetically predisposed individuals are subject to both external and internal factors that elicit the psoriasis initiation and pathogenesis. External risk factors include mechanical damage, individual habits, such as smoking and alcohol consumption, as well as environmental factors like air pollution and sun exposure [48-50].

Psoriasis patients may suffer from various infections due to their weakened immune system. Accordingly, patients who are administered vaccinations against infections experience further exacerbation of their condition. For instance, studies pointed out the possible association between influenza vaccine and psoriasis onset [51,52]. Other researchers found that patients developed psoriasis post-BCG (Bacillus Calmette-Guerin) vaccine shots [53].

Beyond external triggers, internal factors exacerbate the perpetuation of the condition. For instance, people with other medical conditions, including diabetes mellitus, dyslipidemia, and obesity, are at a higher risk of developing psoriasis [54-56]. Hypertension and mental stress are other risk fac-tors, and the latter needs to be extensively investigated to further corroborate its relationship with psoriasis progression. Hypertension, in contrast, was considerably associated with an increased risk of psoriasis [57].».

Corrected

6.

Line 187:  The chapter is not clear; if possible, rewrite it, or add a phase in which you underlie that MSCs have a basic properties in modulating the immune system and the additional ones induced by an inflamed environment (Jiang W, Xu J. Immune modulation by mesenchymal stem cells. Cell Prolif. 2020 Jan;53(1):e12712. doi: 10.1111/cpr.12712. Epub 2019 Nov 15. PMID: 31730279; PMCID: PMC6985662). In this chapter, I would have listed and explain the major mechanisms attributed to MScs in inducing immunosuppressive effects in Psoriasis or in general. As such as, for a complete information, add that although MSCs work essentially by extracellular vesicles (EVs), it is also important to say that these cells may also affect immune cells by directly interaction with them. For example, ICAM-1 expressed by MSCs, is able to interact with LFA-1 on T cells and NK cells and reduce their ability to attack other cells. Inflammatory microenvironment may promote up regulation of these molecules on MSCs. Add this information to have a complete vision of the argument (1) Ren, G. et al. Inflammatory cytokine-induced intercellular adhesion molecule-1 and vascular cell adhesion molecule-1 in mesenchymal stem cells are critical for immunosuppression. J. Immunol. 184, 2321–2328 (2010); 2) Kyurkchiev D, Bochev I, Ivanova-Todorova E, Mourdjeva M, Oreshkova T, Belemezova K, Kyurkchiev S. Secretion of immunoregulatory cytokines by mesenchymal stem cells. World J Stem Cells. 2014; 6(5):552–70. 56).

We agree with your opinion and have substantially revised Chapter 6. «Immunomodulatory capability of MSCs and the role of MSC preconditioning». The literature you suggested has been reviewed and included in the article.

«The immunomodulatory effects of MSCs are achieved through the paracrine production of various growth factors, cytokines and mediators of angiogenesis that enable them to modulate immune responses both directly and indirectly. The secretion of IL-8, IL-6 and granulocyte-macrophage colony-stimulating factor (GM-CSF) by MSCs enhances neutrophil migration to the of infection or injury, thereby increasing their activation and phagocytic capacity, as well as promoting their survival. Additionally, MSCs produce a number of important molecules such as transforming growth factor-beta (TGF-β), hepatocyte growth factor (HGF), prostaglandin E2 (PGE2), soluble human leukocyte an-tigen-G5 (HLA-G5) protein, indoleamine-2,3-dioxygenase (IDO) and induced nitric oxide synthase (iNOS) and IL-10, IL-4, IL-2 to influence the proliferation, differentiation, maturation, and functional activity of various immune cell types, such as T cells, B cells, DCs, macrophages and NK cells. Furthermore, MSCs do not express MHC II, CD40, CD80 and CD86 molecules, which play a role in T cell activation during graft rejection [101]. MSCs express intercellular adhesion molecule-1 (ICAM-1), which can interact with lymphocyte function-associated antigen-1 (LFA-1) on T cells and NK cells, thereby reducing their cytotoxicity and proliferation [102,103 Ren G., 2010; Kyurkchiev, 2014]. MSCs can express non-classical human leukocyte antigen-G (HLA-G), a molecule that plays a role in immunological tolerance. HLA-G expression by MSCs can inhibit the cytotoxic activity of NK cells and cytotoxic T cells and increase the expansion of regulatory T cells (Tregs) [104].

One of the main mechanisms by which human MSCs exert immunosuppressive effects is the production of IDO, which is involved in the degradation of L-tryptophan, leading to its depletion and the accumulation of kynurenine. This in turn inhibits the activation, proliferation and functional activity of T cells, DCs, NK cells and Th17 cells differentiation. Inflammatory diseases stimulate MSC activation by the action of certain cytokines such as IFN-γ, TNF-α, IL-17 and IL-1β. This activation leads to increased ex-pression of MHC class I/II molecules and costimulatory molecules. It also increases the proliferation and survival of these cells and their immunomodulatory and immunosuppressive functions [105].

Therapeutic and immunomodulatory effects of cytokines, growth factors, enzymes, and other molecules are partially derived from the activity of MSCs and partially from their interaction with the inflamed environment. MSCs can produce various cytokines, including TGF-β, HGF, and PGE2, especially in response to inflammatory stimuli such as interferon-gamma (IFN-γ) and TNF-α. Additionally, other cytokines like IL-6, IL-10, and IL-1β are often found in the environment where MSCs are active, and their production can be further enhanced through the crosstalk between MSCs and immune cells. This dynamic interaction between MSCs and the inflamed microenvironment ensures a coordinated production of these molecules, which contribute to the modulation of immune responses and tissue repair processes [106 Jiang, 2020].

……

Preconditioning with IFN-γ or TNF-α can increase IDO synthesis in MSCs. Subsequently, this promotes the differentiation of monocytes into IL-10-secreting M2 macrophages. This M2 macrophages exert immunosuppressive effects. Pretreatment of MSCs with IL-17 inhibits the secretion of T helper 1 (Th1) cells cytokines (TNF-α, IFN-γ and IL-2) by T cells and also promotes the activation of Tregs [108]. Additionally, molecules like ICAM-1 can be upregulated in the presence of an inflamed microenvironment, thereby strengthening the interaction between MSCs and immune cells [102 Ren G., 2010].

MSC-Exo are key mediators of their immunomodulatory effects. Exosomes contain microRNAs (miRNAs), proteins, and lipids that can alter the activity of immune cells, such as reducing the expression of pro-inflammatory cytokines in macrophages and promoting their conversion to an anti-inflammatory phenotype. Through release of exosomes containing immunoregulatory molecules, MSCs can inhibit the activity of pro-inflammatory immune cells and support the function of Tregs [122].

In summary, the immunomodulatory effects of MSCs are a result of both their intrinsic properties and their dynamic response to external inflammatory cues, as well as their ability to influence immune responses through both secreted factors and direct cellular interactions [106]».

Corrected

7.

Line 194: Add also neutrophils.

The comment has been accepted and corrected.

«The immunomodulatory properties of MSCs are due to their effects on the cytokine profile of lymphocytes [99]. They also inhibit the activity of certain immune cell types such as neutrophils, DCs, anti-inflammatory macrophages, natural killer (NK) cells, T- and B-cells, contributing to an overall immunosuppressive environment [100].».

Moreover, in the first paragraph of the Introduction section we also indicated a characteristic feature of psoriasis, in terms of neutrophil infiltration.

«Psoriasis is characterized by hyperproliferative keratinocytes and infiltration of T cells, dendritic cells (DCs), macrophages and neutrophils [1].».

Corrected

8.

Line 204: «Therapeutic and immunomodulatory effects of cytokines, growth factors, enzymes and other molecules produced by MSCs.». Are you sure that MSCs produce all these molecules? Are they molecules present in an inflamed environment? Are molecules produced in part by MSCs and in part by environment? Please correct and clarify.

Your request to expand on this thesis has been accepted and clarified.

«Therapeutic and immunomodulatory effects of cytokines, growth factors, enzymes, and other molecules are partially derived from the activity of MSCs and partially from their interaction with the inflamed environment. MSCs can produce various cytokines, including TGF-β, HGF, and PGE2, especially in response to inflammatory stimuli such as interferon-gamma (IFN-γ) and TNF-α. Additionally, other cytokines like IL-6, IL-10, and IL-1β are often found in the environment where MSCs are active, and their production can be further enhanced through the crosstalk between MSCs and immune cells. This dynamic interaction between MSCs and the inflamed microenvironment ensures a coordinated production of these molecules, which contribute to the modulation of immune responses and tissue repair processes [106 Jiang, 2020].».

Corrected

9.

Line 266:  Add also other information and literature: MSCs from the skin of individuals with psoriasis (pD-MSCs) showed a higher expression of genes responsive for encoding Th1 and Th17 cytockines than normal dermal MSCs nD-MSCs). (Imbalance of Th21-Th17/Th2 in pD - MSCs.). Moreover, nD-MSCs have a more potent inhibitory effect on T cell proliferation than pD- MCSs (Zhao et al. Imminomodulation effect of psoriasis derivated dermal mesenkimal stem cells on Th1/Th17 cell.  Eur. J Dermatol. 31 (3) 2021: 318- 325.). In particular nD-MSCs have greter efficacy compared to pD-MSCs in suppressing the proliferation of CD3 T cells and inducing their apoptosis. These cells can also reduce the Th1/Treg ratio and improve the immunosuppressive function of Tregs.  nD-MCSs can also outperformance pD-MSCs in increasing TGF-b receptor partially restoring the biological immunosuppressive function of psoriatic CD3T cells throught upregulation of TGF-b receptor/signaliing . (J. Jiao, et al., Normal mesenchymal stem cells can improve the abnormal function of T cells in psoriasis via upregulating transforming growth factor-β receptor, J. Dermatol. 49 (10) (2022) 988–997).

«Paragraph 6.1 In vitro studies» has been substantially revised taking into account your comments. The literature you suggested has also been reviewed and added to the article.

«7.1. In vitro studies

In vitro studies investigating the effects of MSCs on psoriasis offer intricate insights into the pathophysiology of this complex dermatological disorder. In the context of psoriasis, the microenvironment exhibits distinct characteristics compared to atopic derma-titis and healthy skin. This specific environment induces resident MSCs to release pro-inflammatory and angiogenic mediators, while simultaneously diminishing their antioxidant capacity, a combination that contributes to the development of psoriatic skin lesions. This highlights the importance of understanding the dynamic interplay between the microenvironment and resident MSC populations in disease progression [123].

Furthermore, psoriatic dermal MSCs (p-DMSCs) reveal a multifaceted role in the pathogenesis of psoriasis. These cells display a remarkable ability to upregulate the ex-pression of vascular endothelial growth factor (VEGF), thereby promoting angiogenesis and facilitating the migration of human umbilical vein endothelial cells (HUVECs). This angiogenic potential suggests a contributory role of p-DMSCs in the enhanced vascularity observed in psoriatic lesions [124]. Additionally, p-DMSCs exhibit a dual effect on keratinocytes, the predominant cell type in the epidermis. They not only promote keratinocyte proliferation but also suppress apoptosis by downregulating caspase-3 ex-pression. Consequently, this dysregulation in keratinocyte dynamics leads to hyperproliferation and impaired apoptosis, exacerbating the inflammatory response within the epidermal layer [125].

Moreover, normal dermal MSCs (n-DMSCs) have been shown to outperform p-DMSCs in suppressing T cell proliferation and enhancing Treg activity. It was found that p-DMSCs have higher expression of genes responsible for the activation of Th1 and Th17 cytokines than n-DMSCs. Compared to p-DMSCs, n-DMSCs not only more effectively inhibit CD3 T cell proliferation but also promote T cell apoptosis, leading to a re-duction in the Th1/Treg ratio. This suggests that n-DMSCs could partially restore immune balance through upregulation of TGF-β receptor signaling pathways, enhancing the immunosuppressive function of Tregs [126,127].

Finally, a study aimed to evaluate the influence of hUC-MSCs on MSCs derived from psoriatic patients (PsO-MSCs) sheds light on the interaction between different MSC populations. hUC-MSCs and PsO-MSCs were isolated and characterized, followed by an indirect co-culture experiment. The effects of co-culture on proliferation and expression of cytokines associated with Th1/Th17 and Th2 pathways were assessed. Intriguingly, the results revealed that prior to co-culture, the proliferation of PsO-MSCs was significantly higher than that of hUC-MSCs, suggesting inherent differences in the proliferative capacities of MSCs derived from psoriatic patients compared to those from healthy sources [128].

Moreover, both unmodified MSCs and those treated with superoxide dismutase 3 (SOD3) exhibit profound inhibition of T cell proliferative responses in vitro. This inhibition underscores the potential of MSC-based therapies in modulating the dysregulated immune responses characteristic of psoriasis, potentially offering a novel avenue for therapeutic intervention [129 Sah, 2016]. Furthermore, recent research highlights that exosomes derived from IFN-γ-stimulated hUC-MSCs exhibit notable immunomodulatory effects. These vesicles, by inhibiting peripheral blood mononuclear cell and T cell proliferation, significantly reduced cytokine production linked to inflammation in vitro [130].

One compelling finding is the capacity of human umbilical cord MSC-derived exosomes (hUCMSCs-Exo) to suppress DCs maturation and activation. This suppression is accompanied by a significant reduction in the expression levels of IL23, a pivotal cytokine implicated in the initiation and perpetuation of psoriatic inflammation [131].

In another study, human umbilical cord blood mononuclear cell-derived exosomes (UCB-MNC-Exo) decreased the expansion of CD4+ and CD8+ T cells, production of cytokines, and promoted an increase in Treg levels in vitro. UCB-MNC-Exo lowered levels of IL-6, IL-8, C-X-C motif chemokine ligand 10 (CXCL10), cyclooxygenase-2 (COX-2), S100A7, and defensin beta 4 (DEFB4), which are markers associated with inflammation and the psoriatic disease process [132].

Overall, these comprehensive in vitro findings delineate the specific interplay be-tween MSCs and various cellular components involved in psoriasis pathogenesis. Such insights provide a solid foundation for the development of targeted therapeutic strategies aimed at ameliorating the symptoms and halting the progression of psoriasis.».

Corrected

10.

Line 355: Table 3: Please add the bibliography and the mechanism of action sequentially in the table with the same sequence exposed in the “In vitro studies chapter”. Bibliography number 85, 86 72 is missing.

The comment has been taken into account and corrected. The references in Table 3 are given sequentially in accordance with the text. Missing data (psoriasis model, animals, age of animals, source of MSC, rote of administration, number of repetitions, etc.) for the following bibliography have been added to the table:

- Chen, Y.; Hu, Y.; Zhou, X.; Zhao, Z.; Yu, Q.; Chen, Z.; Wang, Y.; Xu, P.; Yu, Z.; Guo, C.; Zhang, X.; Shi, Y. Human umbilical cord-derived mesenchymal stem cells ameliorate psoriasis-like dermatitis by suppressing IL-17-producing γδ T cells. Cell Tissue Res. 2022, 388, 549-563. https://doi.org/10.1007/s00441-022-03616-x.

- Ren, X.; Zhong, W.; Li, W.; Tang, M.; Zhang, K.; Zhou, F.; Shi, X.; Wu, J.; Yu, B.; Huang, C.; Chen, X.; Zhang, W. Human umbilical cord-derived mesenchymal stem cells alleviate psoriasis through TNF-α/NF-κB/MMP13 pathway. Inflammation. 2023, 46, 987-1001. https://doi.org/10.1007/s10753-023-01785-7.

- Rodrigues, S.C.; Cardoso, R.M.S.; Freire, P.C.; Gomes, C.F.; Duarte, F.V.; Pires das Neves, R.; Simões-Correia, J. Immunomodulatory properties of umbilical cord blood-derived small extracellular vesicles and their therapeutic potential for inflammatory skin disorders. Int. J. Mol. Sci. 2021, 22, 9797. https://doi.org/10.3390/ijms22189797.

Corrected

11.

Table 4. The references are not reported sequentially as in the chapter "clinical studies" and some are missing 90. Delete the 98.

The comment on Table 4 has been corrected. References are listed sequentially. The missing information has been added to the table (Wang, S.G.; Hsu, N.C.; Wang, S.M.; Wang, F.N. Successful treatment of plaque psoriasis with allogeneic gingival mesenchymal stem cells: a case study. Case Rep. Dermatol. Med. 2020, 2020, 4617520. https://doi.org/10.1155/2020/4617520). Bibliography 98 deleted.

Corrected

12.

Line 430: if you mention that exosome use is more convenient and safe, briefly explain why.

Your comments were taken into account and information was added to the conclusions justifying the advantages of using MSC exosomes.

«.....The therapeutic effect of MSCs can be heterogeneous depending on delivery protocols, MSC populations, and cell drug delivery conditions. MSCs are capable of inducing var-ious immunomodulatory responses that depend on the proinflammatory or immuno-suppressive phenotypes. Other limitations include their ability to spread, lifespan, potential for contamination, and rejection ability. Currently, there are studies that confirm that MSCs achieve a therapeutic effect in vivo mainly due to paracrine signaling through the secretome (through exosomes), rather than regenerative abilities. Thus, the above facts increase the priority of using exosomes compared to MSCs themselves.».

Corrected

Reviewer 2 Report

Comments and Suggestions for Authors

The article “Psoriasis: The versatility of mesenchymal stem cells and exosome therapies” reviews the problem of an actual disease of the contemporary society - psoriasis. Psoriasis is an immune cell-mediated inflammatory skin and joints disease. It can have a substantial negative impact on the physical, emotional and psychological well-being of patients. Psoriasis is widespread in the world, but predominant among certain ethnic groups. It has a strong genetic component, but environmental factors also have a significant role in the spread of the disease. There are several clinical manifestations of psoriasis on the skin, but most often the disease reveals itself in the form of chronic, erythematous, peeling papules and plaques. The recent advancements from preclinical and clinical studies of MSCs, MSC-Exo in the treatment of psoriasis and the mechanisms of therapeutic action are also discussed in the article.

Still, I would like to draw the attention of the authors to some issues.

1. It would be appreciated if the author compared MSCs and exosomes with other methods of psoriasis therapy.

2. Perhaps, the authors could describe Figure 1 in the text in more detail.

3. The mechanism of MSC’s action in psoriasis (Figure 2) might also appreciate a more detailed discussion.

4. Lastly, the are too many abbreviations, making it difficult to understand the manuscript. For example, OMAG (line 143), RCM (line 144) are not used once in the text, and hucMSCs-Exo (line 225) is used just one time.

Comments on the Quality of English Language

Minor editing of the grammar and sentence structure may be required.

Author Response

 Comments for Reviewer 2

Dear Reviewer, thank you for your useful comments and remarks, which will help us improve our work. We have tried to correct all the remarks. In the table you can find our comments and the location of the corrections in the manuscript of the article. Thank you very much for your work!

No.

Reviewer's comment

Reply to comment

Status

1.

It would be appreciated if the author compared MSCs and exosomes with other methods of psoriasis therapy.

Thank you for your comment, we tried to answer it in the article.

In particular, we have added information on psoriasis therapy with MSCs and MSC-exosomes to Table 1. Treatment methods for psoriasis and to the relevant section of the article.

Corrected

2.

Perhaps, the authors could describe Figure 1 in the text in more detail.

Chapter 4 «Treatment methods for psoriasis» has been substantially revised, including Table 1 and Figure 1.

Table 1 has been supplemented with information on psoriasis therapy with MSCs and MSC-Exo, as well as their therapeutic effect.

At the same time, Chapter 3. «Signs and symptoms of psoriasis» is supplemented with more detailed information about the role of genetic factors in the occurrence and progression of

psoriasis.

«Genetic factors play a substantial role in the onset and progression of psoriasis. Certain populations have genetic predispositions that make them more susceptible to developing it. Previous genome-wide studies identified several major susceptibility loci. Among them, the major histocompatibility complex (MHC) region, particularly class I HLA genes like HLA-C, is strongly associated with psoriasis. The HLA-C*06:02 allele is especially significant, being identified as a primary genetic factor in psoriasis susceptibility across different populations [47].

Genetically predisposed individuals are subject to both external and internal factors that elicit the psoriasis initiation and pathogenesis. External risk factors include mechanical damage, individual habits, such as smoking and alcohol consumption, as well as environmental factors like air pollution and sun exposure [48-50].

Psoriasis patients may suffer from various infections due to their weakened immune system. Accordingly, patients who are administered vaccinations against infections experience further exacerbation of their condition. For instance, studies pointed out the possible association between influenza vaccine and psoriasis onset [51,52]. Other researchers found that patients developed psoriasis post-BCG (Bacillus Calmette-Guerin) vaccine shots [53].

Beyond external triggers, internal factors exacerbate the perpetuation of the condition. For instance, people with other medical conditions, including diabetes mellitus, dyslipidemia, and obesity, are at a higher risk of developing psoriasis [54-56]. Hypertension and mental stress are other risk factors, and the latter needs to be extensively investigated to further corroborate its relationship with psoriasis progression. Hypertension, in contrast, was considerably associated with an increased risk of psoriasis [57].».

Corrected

3.

The mechanism of MSC’s action in psoriasis (Figure 2) might also appreciate a more detailed discussion.

Thank you for your comment. It has been taken into account and corrected.

Figure 2 summarizes the new mechanisms of therapeutic action of MSCs and MSC-Exo discovered during preclinical studies in animals. The main points describing these molecular mechanisms are presented in the text of Section 7.2. «In vivo studies», as reported in the article.

«At the same time, preclinical studies in animal models (in vivo) have reported new mechanisms of action of MSCs and MSC-Exo that contribute their therapeutic effect in alleviating the symptoms of psoriasis. These mechanisms of action are summarized in Figure 2 and are also described below in the text.».

We also added one paragraph on the already well-known molecular mechanisms of action of MSCs and MSC-Exo (paragraph 1).

«MSCs and MSC-Exo alleviate psoriasis symptoms through a variety of molecular mechanisms. One key mechanism involves the enhancement of CD4+ T cell activity, which leads to an increased production of IL-4+ and induces apoptosis in cytotoxic T cells. Additionally, MSCs and their exosomes modulate the function of Th1, Th2, Th17 and Treg cells, resulting in elevated levels of IL-10. These actions help maintain an anti-inflammatory environment. The inflammatory environment activates keratinocyte proliferation that further produces inflammatory cytokines (IL-17A, IL-22) and chemokines (CC motif chemokine ligand 20 (CCL20), CCL27). This positive feedback loop could be halted by the inhibitory action of MSCs and their exosomes. Furthermore, these cells can target dendritic cells, macrophages, and other T cells to curb the production of additional inflammatory factors [133].».

Corrected

4.

Lastly, the are too many abbreviations, making it difficult to understand the manuscript. For example, OMAG (line 143), RCM (line 144) are not used once in the text, and hucMSCs-Exo (line 225) is used just one time.

Thanks for the advice. The comment regarding abbreviations has been corrected. The full explanation of the abbreviation is given at its first mention in the article.

Corrected
